# Occupational exposures and exacerbations of asthma and COPD—A general population study

Stinna Skaaby[1]*, Esben Meulengracht Flachs[1], Peter Lange[2,3,4,5], Vivi Schlünssen[6,7], Jacob Louis Marott[4,5], Charlotte Brauer[1], Børge G. Nordestgaard[4,5,8], Steven Sadhra[9], Om Kurmi[10,11], Jens Peter Ellekilde Bonde[1,2]

1 Department of Occupational and Environmental Medicine, Bispebjerg Frederiksberg Hospital, Copenhagen University Hospital, Copenhagen, Denmark, 2 Section of Epidemiology, Institute of Public Health, University of Copenhagen, Copenhagen, Denmark, 3 Department of Medicine, Herlev Gentofte Hospital, Copenhagen University Hospital, Herlev, Denmark, 4 Copenhagen City Heart Study, Bispebjerg Frederiksberg Hospital, Copenhagen University Hospital, Copenhagen, Denmark, 5 Copenhagen General Population Study, Herlev Gentofte Hospital, Copenhagen University Hospital, Herlev, Denmark, 6 Department of Public Health, Environmental, Work and Health, Danish Ramazzini Centre, University of Aarhus, Aarhus, Denmark, 7 National Research Center for the Working Environment, Copenhagen, Denmark, 8 Department of Clinical Biochemistry, Herlev Gentofte Hospital, Copenhagen University Hospital, Herlev, Denmark, 9 Institute of Occupational and Environmental Medicine, College of Medical and Dental Sciences, University of Birmingham, Birmingham, United Kingdom, 10 Faculty of Health and Life Sciences, Coventry University, Coventry, United Kingdom, 11 Division of Respirology, Department of Medicine, McMaster University, Hamilton, Canada

* stinna.skaaby@regionh.dk

**Data Availability Statement:** The combined set of data used in this study can be made available through a trusted third party, Statistics Denmark. Requests for data may be sent to Statistics

## Abstract

### Purpose

Recent studies suggest that occupational inhalant exposures trigger exacerbations of asthma and chronic obstructive pulmonary disease, but findings are conflicting.

### Methods

We included 7,768 individuals with self-reported asthma (n = 3,215) and/or spirometric airflow limitation (forced expiratory volume in 1 second ($FEV_1$)/ forced expiratory volume (FVC) <0.70) (n = 5,275) who participated in The Copenhagen City Heart Study or The Copenhagen General Population Study from 2001–2016. Occupational exposure was assigned by linking job codes with job exposure matrices, and exacerbations were defined by register data on oral corticosteroid treatment, emergency care unit assessment or hospital admission. Associations between occupational inhalant exposure each year of follow-up and exacerbation were assessed by Cox regression with time varying exposure and age as the underlying time scale.

### Results

Participants were followed for a median of 4.6 years (interquartile range, IQR 5.4), during which 870 exacerbations occurred. Exacerbations were not associated with any of the selected exposures (high molecular weight sensitizers, low molecular weight sensitizers,

Denmark: http://www.dst.dk/en/OmDS/
organisation/. Data from the two cohorts, the
Copenhagen City Heart Study and the Copenhagen
General Population Study may be available for
researchers who meet the criteria for access to
confidential data. Contact information can be found
at https://www.frederiksberghospital.dk/afdelinger-
og-klinikker/oesterbroundersoegelsen/kontakt/
Sider/default.aspx#10 and https://www.
herlevhospital.dk/afdelinger-og-klinikker/klinisk-
biokemisk-afdeling/forskning/Sider/Herlev-
oesterbroundersoegelsen.aspx.

**Funding:** JPB, grant number 40-2016-09
20165103813, The Danish Working Environment
Research Fund https://amff.dk/. The funders had
no role in study design, data collection and
analysis, decision to publish, or preparation of the
manuscript.

**Competing interests:** The authors have declared
that no competing interests exist.

irritants or low and high levels of mineral dust, biological dust, gases & fumes or the composite variable vapours, gases, dusts or fumes). Hazards ratios ranged from 0.8 (95% confidence interval: 0.7;1.0) to 1.2 (95% confidence interval: 0.9;1.7).

## Conclusion

Exacerbations of obstructive airway disease were not associated with occupational inhalant exposures assigned by a job exposure matrix. Further studies with alternative exposure assessment are warranted.

## Introduction

Globally, asthma and chronic obstructive pulmonary disease (COPD) are highly prevalent and common causes of morbidity and mortality [1–3]. While airflow limitation and inflammation in asthma may resolve spontaneously or in response to medication, airway obstruction in COPD is, by definition, persistent. Asthma involves the large and small airways, whereas COPD is a disease primarily in the small airways. The two conditions are overlapping. Patients with asthma might develop chronic airway obstruction, and elements of reversible airflow limitation are often present in COPD [4–6].

Exacerbations are acute worsening of asthma or COPD and are often defined on the basis of management: treatment with oral corticosteroids and antibiotics in an outpatient setting (moderate exacerbations), or managed in emergency care with or without hospital admission (severe exacerbations) [7–9]. Exacerbations are associated with an accelerated loss of lung function among some asthmatic patients [10] and decreased survival in patients with COPD [11, 12]. Possible triggers of exacerbations of asthma and COPD include infections, low temperatures and exposure to different types of airborne particles [13, 14]. Airborne particles include ambient air pollution with well-described associations to exacerbations of COPD [15] and asthma [16–18], and occupational inhalant exposures with much less evident associations. Occupational studies have largely focused on new-onset asthma or COPD [19–22]. It is, however, possible that workplace hazards are associated with exacerbations of asthma and COPD, and that these may cause greater morbidity [23]. Exacerbations of both diseases might be associated with the same inhalant hazards at work but are rarely studied together. Recent studies suggest that different types of inhalant exposures in the workplace are associated with exacerbations of asthma [24] and COPD [25], but rely on self-reported exacerbations which are prone to recall bias. Updated information on the risk of exacerbations is important for evidence-based guidance of asthma and COPD patients in general.

We studied the association between concurrent inhalant occupational exposures and exacerbations of asthma and/or COPD.

## Methods

### Population

Participants were selected from two large cohort studies: The Copenhagen City Heart Study (CCHS) [26] and The Copenhagen General Population Study (CGPS) [27]. CCHS was initiated in 1976, and the fifth round of follow up was completed in 2015. CGPS started in 2003 and is a prospective cohort study with ongoing recruitment of participants. Individuals from the fourth (2001–2003) and/or the fifth (2011–2015) follow up round of CCHS and from

2003–2016 in the CGPS were eligible for the present study. Participants in both studies were 20–100 years old and had been randomly selected from the general population through the Danish Civil Registration Service. All participants gave written informed consent, and both studies were approved by the Danish Ethics Committees. All data were fully anonymized before assessment. At each round of examination, participants filled out a questionnaire, and completed a physical examination at a test center located at a public hospital in Copenhagen. The questionnaire was self-administered, concerning health status, lifestyle and socio-economic status, and was assessed by one of the investigators on the day of attendance. The physical examination included spirometry. Pre-bronchodilator forced expiratory volume in 1 second ($FEV_1$) and forced vital capacity (FVC) were measured by investigators and repeated three times with the participant in a standing position. The test was redone if the two closest trials differed by more than 5%, or the visual appearance of the spirometry tracing was unsatisfactory. A Vitalograph spirometer (Maids Moreton, Buckinghamshire) was used in The Copenhagen City Heart Study and by the first 14,624 individuals in the Copenhagen General Population Study, while an EasyOne Diagnostic Spirometer (ndd Medizintechnik, Switzerland) measured lung function in the remaining individuals.

Individuals were included in the present study, if they reported asthma in the questionnaire and/or had spirometry indicating airflow limitation ($FEV_1$/ FVC < 0.70). Other inclusion criteria in the present study were age 30–60 years at baseline, employment at least one year during the study period, and complete data regarding smoking habits, education, weight, height and spirometry.

A sample of individuals with no reported asthma and with $FEV_1$/FVC $\geq$ 0.70 was constructed to test for differences in baseline exposure. A one-to-three matching was conducted based on age at inclusion, sex, smoking status (never, former, current smoker), BMI category (<18.5, 18.5–24.9, 25–29.9, 30+ kg/m2), education (elementary, high school, academic) and participation after the year 2000.

## Exposure

We combined job codes from the Danish Occupational Cohort database (DOC*X) [28] with job exposure matrices to determine exposure each year of the follow-up period (S1 Table). DOC*X is a database with annual job titles according to the Danish version of the International Standard Classification of Occupation (DISCO-88) on all Danish wage earners from 1970 until present. For the participants with complete job histories, exposure status was relatively stable during employed year of follow up. In case of missing job codes in employed years, prior job titles maximally five years prior were extrapolated. We applied parts of two expert-rated job exposure matrices; the Airborne Chemical Job Exposure Matrix (ACEJEM) [29] commonly used for chronic obstructive lung disease, and the Occupational Asthma-specific JEM (OAsJEM) [30] designed for occupational asthma. The ACE JEM was developed for the UK SOC 2000 classification job codes, the OAsJEM for the International Standard Classification of Occupation (ISCO-88), and both were converted into DISCO-88 codes. The ACE JEM included information on 12 pollutant types (including composites) and assigned proportion of exposed workers (<5%, 5–19%, 20–49%, $\geq$50%), level of exposure (not exposed, low, medium, high) and a binary variable (non-exposed, exposed) to each job code. The OAsJEM covered 30 different sensitizers or irritants, and each job code was classified in three categories: high ("at least 50% of the workers exposed and moderate to high intensity"), medium ("low to moderate probability or low intensity of exposure, such as 'high probability and low intensity' or 'low probability and moderate to high intensity'") and not exposed ("unlikely to be exposed with low probability and low intensity").

To achieve adequate power we selected the following main types of exposure: mineral dust, biological dust, gases & fumes and the composite variable of vapours, gases, dusts or fumes (VGDF) from the ACE JEM, and high molecular weight sensitizers, low molecular weight sensitizer and irritants from the OAsJEM. Probability and intensity of exposure assigned by the ACEJEM were combined into the following classes: no, low and high exposure (S2 Table). Exposure in the OAsJEM was dichotomized into exposed (including high and medium exposure assigned by the OAsJEM) and unexposed.

## Outcome

Exacerbations were defined by treatment with oral corticosteroids, emergency care unit assessment (emergency care) or hospital admission related to asthma or COPD. Cases were identified through linkage with The Danish National Prescription Registry [31] and The Danish National Patient Register [32]. Treatment with oral corticosteroids included prescriptions for prednisolone (ATC code H02AB06) or prednisone (H02AB07). Emergency care or hospital admissions comprised of the following: (1) primary diagnosis "chronic obstructive pulmonary disease" (ICD-code J44) and secondary diagnosis "pneumonia" (J13 to J18) or (2) primary diagnosis "asthma" (J45) or "status asthmaticus" (J46) or (3) primary diagnosis "respiratory failure" (J96) in combination with a secondary diagnosis "chronic obstructive pulmonary disease" (J44) or "asthma" (J45) or "status asthmaticus" (J46). The highest level of treatment per episode was recorded, and the date of prescription, emergency care or hospital admission day denoted an event. Exacerbations one year prior to inclusion were recorded separately. In case of an exacerbation occurring before inclusion and less than four weeks before an event in the follow-up period, the event was regarded as an exacerbation in the previous year.

## Covariates

Based upon status at inclusion, the following covariates were included; sex, smoking status (never, former, current smoker), BMI category ($<18.5$, $18.5–24.9$, $25–29.9$, $30+$ kg/m$^2$), education (elementary, high school, academic), FEV$_1$% predicted class ($<80\%$ and $\geq80\%$) and exacerbations one year prior to study inclusion (none, $\geq1$). Calculation of FEV$_1$% predicted has previously been described [33].

## Statistics

In a follow-up design, we used Cox regression with time-varying exposure to examine the hazard ratio (HR) of exacerbation according to inhalant exposure. Age was the underlying time scale, and end of follow-up was the first occurring exacerbation, exit from the labour market, death or year 2017, whichever came first. We found no interactions between the effects of exposure and sex, exposure and smoking status, exposure and FEV$_1$% predicted or exposure and exacerbations one year prior to inclusion. Stratifying by exacerbation within the year before inclusion or excluding the covariate from the model did not change main findings. We conducted sensitivity analyses including only self-reported asthma, FEV$_1$/FVC$<0.70$ or individuals with a complete job history. To ensure temporality between exposure and outcome we repeated all analyses with inhalant exposure assigned the previous year of all follow-up years. Collinearity of exposures did not allow for analyses including more than one type of exposure in a model. Proportional hazards assumptions were evaluated graphically. SAS version 9.4 (SAS Institute Inc., Cary, NC, USA) was used for statistical analyses. P-values were two-sided, and statistical significance was defined as p$<0.05$.

## Results

A total of 7,768 individuals with self-reported asthma, $FEV_1/FVC < 0.70$ or a combination of the two were included. The mean age at study inclusion was 50 years (standard deviation, SD 7), and 62% were current or former smokers (Table 1).

Exposure to the selected inhalant agents at study inclusion varied from 28% exposed to low levels of vapours, gases, dusts or fumes (VGDF) to 2% exposed to high levels of biological dust and gases & fumes (Table 2). At the time of study inclusion, 61% of the population (N = 4,736) was not exposed to any of the selected inhalant agents. Proportions of exposed in the matched population with no self-reported asthma and $FEV_1/FVC \geq 0.70$ resembled our population (S3 Table).

First time exacerbation since study inclusion was recorded in 870 individuals during a median of 4.6 years (interquartile range, IQR 5.4). The number of exacerbations was 411 among individuals with self-reported asthma only, 317 in the group of participants with $FEV_1/FVC < 0.70$ only, and 142 in the remaining participants. Only 8% of the exacerbations involved emergency care or hospital admission. Exacerbations were associated with low $FEV_1$ at inclusion; HR 1.5 (95% confidence interval [CI] 1.3;1.8), a body mass index above normal at

**Table 1. Characteristics of the study population at inclusion.**

|  | N = 7,768 |
|---|---|
| Age, years, mean (SD) | 50 (7) |
| Sex, male, n(%) |  |
| Male | 3,361 (43) |
| Female | 4,407 (57) |
| BMI, n(%) |  |
| <18.5 | 54 (1) |
| 18.5–24.9 | 3,716 (48) |
| 25–29.9 | 2,869 (37) |
| ≥30 | 1,129 (15) |
| Education, n(%) |  |
| Elementary | 672 (9) |
| High school | 4,774 (61) |
| Academic | 2,322 (30) |
| Smoking, n(%) |  |
| Never smoker | 2,984 (38) |
| Former smoker | 3,083 (40) |
| Current smoker | 1,701 (22) |
| Self-reported asthma, n(%) | 3,215 (42) |
| $FEV_1/FVC < 0.70$, n(%) | 5,275 (68) |
| Self-reported asthma and FEV1/FVC <0.70, n(%) | 722 (9) |
| $FEV_1$% predicted, n(%) |  |
| ≥80% | 5,806 (75) |
| <80% | 1,962 (25) |
| Exacerbations one year prior to inclusion, n(%) |  |
| No | 7,562 (97) |
| ≥ 1 | 206 (3) |

Abbreviations; SD: standard deviation; n: number; BMI: body mass index; $FEV_1$: forced expiratory volume in 1 second; FVC: forced vital capacity.

**Table 2. Exposures at study inclusion.**

| | Exposure, number (row-%) | | |
|---|---|---|---|
| | **Unexposed** | **Low** | **High** |
| **ACE JEM** | | | |
| Vapors, gases, dusts or fumes | 4,906 (63) | 2,184 (28) | 678 (9) |
| Mineral dusts | 6,167 (79) | 1,189 (15) | 412 (5) |
| Biological dusts | 6,368 (82) | 1,276 (16) | 124 (2) |
| Gases&fumes | 7,236 (93) | 352 (5) | 180 (2) |
| | **Unexposed** | **Exposed** | |
| **OAsJEM** | | | |
| High molecular weight sensitizer | 6,739 (87) | 1,029 (13) | |
| Low molecular weight sensitizer | 6,633 (85) | 1,135 (15) | |
| Irritants | 5,889 (76) | 1,879 (24) | |

Abbreviations: ACE JEM: The Airborne Chemical Job Exposure Matrix; OAsJEM: The Occupational Asthma-specific JEM

inclusion; HR for BMI 25–29.9: 1.3 (95% CI: 1.1;1.5); HR for BMI$\geq$30: 1.5 (95% CI 1.3;1.9) and female sex; HR 1.5 (95% CI 1.3;1.8) (S4 Table). Having had an exacerbation in the year before inclusion (n = 206) was associated with a hazard ratio of 6.9 (95% CI 5.6;8.5) of a new exacerbation.

Main results are presented in Table 3. We found no associations between exacerbations and mineral dust, biological dust, gases & fumes, vapours, gases, dusts or fumes (VGDF), high molecular weight sensitizer (HMW), low molecular weight sensitizer (LMW) or irritants. Analyses on self-reported asthma only (S5 Table) or $FEV_1/FVC < 0.70$ (S6 Table) showed similar results except for exposure to low levels of gases & fumes which was associated with a hazard ratio of 1.6 (95% CI 1.1;2.3). Repeating the analyses with exposure assigned one year prior, excluding $FEV_1$% predicted as a covariate or only including individuals with a complete job history did not change our main findings.

## Discussion

Our study is the first to comprehensively assess the association between exacerbations and inhalant occupational hazards in a large population of individuals from the general population with self-reported or spirometric measures indicating asthma or COPD. An exacerbation was recorded in 870 out of 7,768 individuals with self-reported asthma and/or airflow limitation during a median follow-up of 4.6 years (interquartile range, IQR 5.4). In line with findings from clinical cohorts of patients with asthma and COPD, the exacerbation risk was significantly higher in individuals with low lung function and a history of previous exacerbations. There was no association between occupational inhalant exposures and exacerbations. Including only individuals with self-reported asthma or participants with airflow limitation did not alter the results, apart from the observation that low levels of gases & fumes were associated with exacerbations in individuals with self-reported asthma.

The strong association between prior exacerbations and future events is well-established [34, 35]. In our population of individuals with self-reported asthma, 4% exacerbated within the first 12 months of follow-up, and 6% of these were defined by a hospital admission or emergency care. In line with this, a large study of patients with asthma with similar ages and access to health care who received at least one type of asthma medication reported that within 12 months 8% exacerbated and 16% of these required hospital admissions or emergency care in

**Table 3. Associations between inhalant exposures and exacerbations.**

|  | Exacerbations | Follow-up years | Crude | Adjusted* |
|---|---|---|---|---|
|  | *Number* | *Number* | *HR (95% CI)* | *HR (95% CI)* |
| Vapors, gases, dusts or fumes |  |  |  |  |
| No | 553 | 26.340 | 1 (ref) | 1 (ref) |
| Low | 222 | 11.683 | 0.9 (0.8;1.1) | 1.0 (0.8;1.1) |
| High | 78 | 3.508 | 1.1 (0.9;1.4) | 1.0 (0.8;1.3) |
| Mineral dusts |  |  |  |  |
| No | 692 | 33.244 | 1 (ref) | 1 (ref) |
| Low | 114 | 6.133 | 0.9(0.8;1.1) | 1.0 (0.8;1.2) |
| High | 47 | 2.154 | 1.1(0.8;1.4) | 1.0 (0.7;1.3) |
| Biological dusts |  |  |  |  |
| No | 709 | 34.031 | 1 (ref) | 1 (ref) |
| Low | 132 | 6.841 | 0.9 (0.8;1.1) | 0.9 (0.7;1.0) |
| High | 12 | 660 | 0.9 (0.5;1.6) | 0.8 (0.5;1.5) |
| Gases&fumes |  |  |  |  |
| No | 792 | 38.811 | 1 (ref) | 1 (ref) |
| Low | 42 | 1.706 | 1.1 (0.8;1.5) | 1.2 (0.9;1.7) |
| High | 19 | 1.015 | 1.0 (0.7;1.5) | 0.9 (0.5;1.4) |
| High molecular weight sensitizer |  |  |  |  |
| Unexposed | 747 | 35.978 | 1 (ref) | 1 (ref) |
| Exposed | 106 | 5.554 | 0.9 (0.8;1.1) | 0.8 (0.7;1.0) |
| Low molecular weight sensitizer |  |  |  |  |
| Unexposed | 723 | 35.619 | 1 (ref) | 1 (ref) |
| Exposed | 130 | 5.913 | 1.1 (0.9;1.3) | 1.0 (0.8;1.2) |
| Irritants |  |  |  |  |
| Unexposed | 632 | 31.861 | 1 (ref) | 1 (ref) |
| Exposed | 221 | 9.671 | **1.1 (1.0;1.3)** | 1.0 (0.9;1.2) |

Cox regression with time varying exposure and age as underlying time scale *Adjusted for sex, education, smoking status, body mass index and $FEV_1$% predicted.
Abbreviations; HR: hazard ratio; CI: confidence interval.

the UK [34]. A possible explanation for the slightly lower occurrence in our study is that our definition of asthma did not require the use of asthma medication thereby including milder and inactive cases.

Exacerbations of asthma and COPD have been studied separately in recent occupational studies, and results of one study are partly in agreement with our findings [24], whereas others are not [25, 36]. Consistent with our results, JEM assigned exposure to agents with high molecular weight, low molecular weight or irritating properties were not associated with exacerbations treated by oral corticosteroids or requiring emergency treatment or hospital admission [24]. Self-reported exposure to biological dust and the composite variable gas, smoke or dust but not mineral dust was, however, positively associated with exacerbations requiring emergency care treatment or hospital admission, but not to exacerbations controlled by corticosteroids alone. In another study, asthma exacerbations were associated with high and low levels of biological dust and high and not low levels of mineral dust, gases and fumes and a composite variable [36]. In a population of current or former smokers with COPD, intermediate/high risk of exposure in the longest-held job was associated with exacerbations requiring health care utilization with low risk of exposure as a reference [25].

The diverging results might overall be explained by different ways of assessing exacerbations and exposure or the chosen covariates. In all studies mentioned above, exacerbations were self-reported and thereby susceptible to recall bias. Exposure was accounted for differently; not required to be concurrent with exacerbations [25] or the reported significant findings were based on self-reports [24]. We adjusted for body mass index (BMI) and education as a proxy of socioeconomic position. Both have been shown to be directly or indirectly associated with exacerbations of asthma [37–39] and possibly correlated with occupational exposure. The two studies concerning exacerbations of asthma [24, 36] did not control for these which might contribute to the different findings.

Our results suggest that exposure to the selected inhalant hazards is not associated with exacerbations in individuals with airway obstruction who are able to continue to work. Improved technology and governmental regulation are important contributors to a large decrease in most occupational inhalant exposures since the 1970s [40] making findings plausible. Traditionally, asthma and COPD have not been studied together in the occupational setting. However, the diseases are overlapping and difficult to distinguish between solely based on data available in our cohorts. Even in studies with access to post-bronchodilator pulmonary function data, reversibility was found in 44–50% of patients with COPD [41, 42], and 25% of asthma patients aged 55 or older had a co-existent diagnosis of COPD [43]. In analyses restricting the population to self-reported asthmatics, we found that low levels of gases & fumes were associated with exacerbations with a hazard ratio of 1.6 (1.1;2.3). The finding might be explained by multiple testing, but is biologically plausible, as asthma exacerbations are also associated with outdoor ambient particulate matter [44]. Regardless, our finding needs to be replicated in other studies.

Strengths of the study included register-based job titles year by year, securing concurrent exposure. Exacerbations were identified in registers and not prone to recall bias. The population represented a wide range of the general population enabling analyses of exposed or unexposed individuals with the same educational level as a proxy of socioeconomic position. Exposure rates were comparable to a matched group of controls.

Our study has limitations. The population was selected by a self-reported diagnosis of asthma or spirometry indicating airflow limitation. A large proportion of individuals with $FEV_1/FVC$ below 0.70 was never smokers in the present study. Some of these may have undiagnosed asthma. However, a study with post-bronchodilator spirometry reported similar findings among never smokers [45]. In total, 312 exacerbations occurred among individuals with $FEV_1/FVC<0.70$ and no self-reported asthma, and 24% (74/312) of these among never smokers suggesting that this group of individuals were indeed ill. Exposure was assigned by job exposure matrices (JEM), which inevitably causes misclassification, as JEMS do not account for variations in levels of exposure within jobs or at the individual level. However, if the mean exposure level for a given job is accurate, this misclassification is not likely to result in attenuated risk estimates, because the measurement error is of Berkson type [46]. We do acknowledge that validation studies for the applied JEMS are not available, and therefore non-differential misclassification towards zero cannot be ruled out. The occupational airborne chemical exposure matrix (ACE JEM) [29] and the occupational asthma-specific JEM (OAs-JEM) [30] were created with an emphasis on detecting new-onset asthma and COPD rather than exacerbations. The selected categories of exposure were, however, considered to be possible occupational triggers of exacerbations of COPD and asthma. We were not able to account for the use of respiratory protective equipment (RPE), as this was not included in the ACE JEM or in the questionnaire. Legislation in Europe introduced in the 1980s has focused on adjustment of use of RPE as well as assessing its effectiveness, and thus RPE is now considered a last resort of protection. Exacerbations were identified by prescription for oral

corticosteroids, which are also prescribed for other diseases such as rheumatoid arthritis and inflammatory bowel disease. Yet, the method has previously been validated and is generally accepted [47], and the risk of bias is considered non-differential. Finally, we did not control for ambient air pollution, as our population was urban. Our population was relatively young, and we did not adjust for comorbid disease. We did not have information on atopy, which may play a role in asthma exacerbations, but its role in late-onset asthma is considered small [48]. Our study population is not representative of all patients with airflow limitations. The mean age at inclusion was 50 years old, and the median follow-up time was 4.6 years. Traditionally, COPD was considered a disease of those aged >50 years, but is suggested to be detectable in 20–45 year old individuals [49]. Still, our population is young. As concomitant exposure was essential in our study, we did not include older participants. Only 9% of the participants reported elementary school as highest level of education. The corresponding rate in the general population aged 35–65 years old in the capital region of Denmark in 2008 was 21% [50] and 24% in the first round of examinations in The Copenhagen General Population Study. A possible explanation for the lower frequency in our population is that the overall lower employment rates among individuals with asthma and COPD are most pronounced in lower educational levels [51–53]. Consequently, power in the present study may be affected.

In conclusion, our results indicate that occupational exposures in Danish individuals who continue to work despite asthma and COPD are not associated with a higher risk of exacerbations.

## Supporting information

**S1 Table. Overview of the methodology.**
(DOCX)

**S2 Table. Exposure classes combining level and proportion of exposure assigned by the Airborne Chemical Job Exposure Matrix.**
(DOCX)

**S3 Table. Exposure at study inclusion in study population and matched group.**
(DOCX)

**S4 Table. Full Cox regression model with time varying exposure and age as underlying time scale.**
(DOCX)

**S5 Table. Associations between exposure and exacerbations of self-reported asthma.**
(DOCX)

**S6 Table. Associations between selected inhalant hazards and exacerbations in individuals with FEV1/FVC<0.70.**
(DOCX)

## Acknowledgments

We would like to thank the authors of the OAsJEM for providing us with the matrix.

## Author Contributions

**Formal analysis:** Stinna Skaaby, Esben Meulengracht Flachs, Jens Peter Ellekilde Bonde.

**Funding acquisition:** Stinna Skaaby, Jens Peter Ellekilde Bonde.

**Methodology:** Stinna Skaaby, Esben Meulengracht Flachs, Peter Lange, Vivi Schlünssen, Jacob Louis Marott, Charlotte Brauer, Børge G. Nordestgaard, Steven Sadhra, Om Kurmi, Jens Peter Ellekilde Bonde.

**Software:** Steven Sadhra, Om Kurmi.

**Writing – original draft:** Stinna Skaaby.

**Writing – review & editing:** Stinna Skaaby, Esben Meulengracht Flachs, Peter Lange, Vivi Schlünssen, Jacob Louis Marott, Charlotte Brauer, Børge G. Nordestgaard, Steven Sadhra, Om Kurmi, Jens Peter Ellekilde Bonde.

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
