## [Decision Letter · Decision Letter 0]

19 Nov 2020

PONE-D-20-33101

Occupational exposures and exacerbations of asthma and COPD – a general population study

PLOS ONE

Dear Dr. Skaaby,

Thank you for submitting your manuscript to PLOS ONE. After careful consideration, we feel that it has merit but does not fully meet PLOS ONE’s publication criteria as it currently stands. Therefore, we invite you to submit a revised version of the manuscript that addresses the points raised during the review process.

Please provide revisions and/or explanations for the points suggested by the reviewer.

We look forward to receiving your revised manuscript.

Kind regards,

Davor Plavec, MD, MSc, PhD, Prof.

Academic Editor

PLOS ONE

Additional Editor Comments:

Please provide revisions and/or explanations for the points suggested by the reviewer.

Journal Requirements:

2. In your ethics statement in the manuscript and in the online submission form, please provide additional information about the patient records used in your retrospective study. Specifically, please ensure that you have discussed whether all data were fully anonymized before you accessed them.

Reviewers' comments:

Reviewer's Responses to Questions

**Comments to the Author**

1. Is the manuscript technically sound, and do the data support the conclusions?

Reviewer #1: Yes

2. Has the statistical analysis been performed appropriately and rigorously? 

Reviewer #1: Yes

3. Have the authors made all data underlying the findings in their manuscript fully available?

Reviewer #1: No

4. Is the manuscript presented in an intelligible fashion and written in standard English?

Reviewer #1: Yes

5. Review Comments to the Author

Reviewer #1: Thank you for the opportunity to review this paper. It is very well written, and it addresses the very important topic. The authors have invested a lot of knowledge and attention to the matter. This results will contribute to the growing knowledge in the field.

There are just a few issues I would like to encourage the authors to address:

1. Describe, in general, the questionnaire used.

2. In general, Methods section needs some additional information to be more comprehensible. Lines 61-62 – at which point was this measured? „Pre-bronchodilator FEV1 and FVC were measured at study inclusion and 62 repeated three times with the participant in a standing position.“ It’s unclear whether the authors have done it (i.e. approached these respondents and performed spirometry) or was it a question in questionnaire). If the respondents were approached, describe how did you approach the respondents and where did the examinations take place.

3. A flowchart of the procedure would be out most helpful to the readers. What came first? Census – cohort study –Which occupations – the exposed group – ACEJEM / OAsJEM – statistical analysis… etc.

4. Discussion –your angle on “the large decrease in most occupational inhalant exposures since the 1970s (40) combined with more safety equipment”… is sound and interesting. Please provide a short further explanation about PPE and other preventive measures in this subgroup (exposed to hazards). Which technical procedures and PPE are mandatory in Denmark regarding the selected hazards in your study? Were the data on PPE included in the questionnaire?

5. Reference 50 – seems to be incomplete

6. PLOS authors have the option to publish the peer review history of their article (what does this mean?). If published, this will include your full peer review and any attached files.

Reviewer #1: No

---

## [Author Response · Author response to Decision Letter 0]

26 Nov 2020

To editor - please find attached a new cover letter addressing sharing of data. We have updated the paper to hopefully fulfil PLOS ONE's requirements on style and ethics statement.

To reviewers - Please find attached the document entitled Response to Reviewers.

---

## [Editor Report · Decision Letter 1]

27 Nov 2020

Occupational exposures and exacerbations of asthma and COPD – a general population study

PONE-D-20-33101R1

Dear Dr. Skaaby,

We’re pleased to inform you that your manuscript has been judged scientifically suitable for publication and will be formally accepted for publication once it meets all outstanding technical requirements.

Kind regards,

Davor Plavec, MD, MSc, PhD, Prof.

Academic Editor

PLOS ONE

Additional Editor Comments (optional):

With the revisions made the manuscript is ready for publication.
---

## [Editor Report · Acceptance letter]

15 Dec 2020

PONE-D-20-33101R1 

Occupational exposures and exacerbations of asthma and COPD – a general population study 

Dear Dr. Skaaby:

I'm pleased to inform you that your manuscript has been deemed suitable for publication in PLOS ONE. Congratulations! Your manuscript is now with our production department. 

Kind regards, 

on behalf of

Dr. Davor Plavec 

Academic Editor

PLOS ONE